# Metabolic Responses to Arsenite Exposure Regulated through Histidine Kinases PhoR and AioS in *Agrobacterium tumefaciens* 5A

**DOI:** 10.3390/microorganisms8091339

**Published:** 2020-09-02

**Authors:** Rachel A. Rawle, Monika Tokmina-Lukaszewska, Zunji Shi, Yoon-Suk Kang, Brian P. Tripet, Fang Dang, Gejiao Wang, Timothy R. McDermott, Valerie Copie, Brian Bothner

**Affiliations:** 1Department of Microbiology and Immunology, Montana State University, Bozeman, MT 59717, USA; rachel.rawle@gmail.com; 2Department of Chemistry and Biochemistry, Montana State University, Bozeman, MT 59717, USA; tokminalukas@montana.edu (M.T.-L.); brian.tripet@gmail.com (B.P.T.); ffang@montana.edu (F.D.); 3State Key Laboratory of Agricultural Microbiology, College of Life Science and Technology, Huazhong Agricultural University, Wuhan 430070, China; shizunji@gmail.com (Z.S.); gejiao@mail.hzau.edu.cn (G.W.); 4Department of Land Resources and Environmental Sciences, Montana State University, Bozeman, MT 59717, USA; yoonsuksamm@gmail.com (Y.-S.K.); timmcder@exchange.montana.edu (T.R.M.)

**Keywords:** arsenic, arsenite oxidation, metabolomics, NMR, mass spectrometry, multi-omics

## Abstract

Arsenite (As^III^) oxidation is a microbially-catalyzed transformation that directly impacts arsenic toxicity, bioaccumulation, and bioavailability in environmental systems. The genes for As^III^ oxidation (*aio*) encode a periplasmic As^III^ sensor AioX, transmembrane histidine kinase AioS, and cognate regulatory partner AioR, which control expression of the As^III^ oxidase AioBA. The *aio* genes are under ultimate control of the phosphate stress response via histidine kinase PhoR. To better understand the cell-wide impacts exerted by these key histidine kinases, we employed ^1^H nuclear magnetic resonance (^1^H NMR) and liquid chromatography-coupled mass spectrometry (LC-MS) metabolomics to characterize the metabolic profiles of Δ*phoR* and Δ*aioS* mutants of *Agrobacterium tumefaciens* 5A during As^III^ oxidation. The data reveals a smaller group of metabolites impacted by the Δ*aioS* mutation, including hypoxanthine and various maltose derivatives, while a larger impact is observed for the Δ*phoR* mutation, influencing betaine, glutamate, and different sugars. The metabolomics data were integrated with previously published transcriptomics analyses to detail pathways perturbed during As^III^ oxidation and those modulated by PhoR and/or AioS. The results highlight considerable disruptions in central carbon metabolism in the Δ*phoR* mutant. These data provide a detailed map of the metabolic impacts of As^III^, PhoR, and/or AioS, and inform current paradigms concerning arsenic–microbe interactions and nutrient cycling in contaminated environments.

## 1. Introduction

Arsenic is the highest priority EPA contaminant due to its prevalence, toxicity, and potential for wide-spread human exposure [1]. Contamination of water and soil systems across the world has led to over 200 million human exposures and is associated with a variety of diseases and cancers [2,3]. The toxicity and bioavailability of arsenic is directly related to its chemical speciation, and in virtually all environments studied it is well established that microbes are the principal drivers of this speciation [4]. Thus, understanding bacterial arsenic speciation events, how they are regulated, and their associated metabolic effects are essential for addressing environmental arsenic contamination.

Arsenite (As^III^) oxidation is an important chemical transformation, during which the more toxic As^III^ species is oxidized to less toxic arsenate (As^V^). *Agrobacterium tumefaciens* 5A is a model organism for As^III^ oxidation and research on this organism during the past decade has revealed several key features about the control of bacterial As^III^ oxidation. The As^III^ oxidase (AioBA) is regulated by a three-component signal transduction system: periplasmic As^III^ sensor protein (AioX), histidine kinase (AioS), and its cognate regulatory partner (AioR) [5,6]. As^III^ is sensed in the periplasm by AioX, which then transfers this signal to AioS; AioS phosphorylates AioR, which in turn induces expression of the As^III^ oxidase. As^III^ is subsequently oxidized in the periplasm and the resulting As^V^ (a phosphate analog) can enter the cytoplasm via phosphate transporters. Recent studies have identified important regulatory links between the phosphate stress response (PSR) and As^III^ oxidation [7,8]. The PSR is regulated through a two-component signal transduction system (PhoR/PhoB), where the histidine kinase PhoR is the master regulator controlling expression of *aioSRBA* [8], in addition to the well-defined PSR genes [9,10]. Cross talk between these two regulatory pairs, PhoR/PhoB and AioS/AioR, has also been demonstrated [8]. Improved growth under low-Pi conditions following As^III^ exposure and evidence for partial incorporation of As^V^ into cellular lipids in *A. tumefaciens* 5A [8] indicate a close relationship between these regulatory components.

Recent transcriptomics experiments on *A. tumefaciens* 5A wild-type, Δ*phoR,* and Δ*aioS* strains reported that As^III^ exposure induces global cell responses, many of which involve PhoR and to a lesser extent, AioS [11]. These data have expanded the traditional view of arsenic impacts to one that now involves multiple fundamental nutrient cycles. In addition to arsenic resistance and oxidative stress responses, carbon metabolism, iron metabolism, and various transport systems are affected. Additionally, initial metabolomics experiments on wild-type *A. tumefaciens* 5A reported significant metabolic changes during As^III^ exposure and revealed key disruptions in central carbon metabolism [12]. Together, these studies have laid the foundation for a comprehensive understanding of arsenic exposure in As^III^-oxidizing bacteria. While metabolomics analysis was performed on wild-type *A. tumefaciens* 5A cells, the metabolic adaptations controlled by PhoR and AioS under As^III^ exposure remained poorly understood. We have employed a global metabolomics approach using liquid chromatography-coupled mass spectrometry (LC-MS) and ^1^H nuclear magnetic resonance (NMR) spectroscopy to assess the metabolic adaptations of Δ*phoR* and Δ*aioS* mutants during As^III^ exposure and oxidation. Specifically, we aimed to characterize cellular metabolome changes that result from the disruption of PhoR and/or AioS signaling. The metabolomics data generated provide a direct read-out of metabolic networks impacted by the regulatory activities of these histidine kinases. In addition, we assimilated this work with our previous metabolomics and transcriptomics studies on wild type *A. tumefaciens* [11,12] to put forth a current, multi-omics model of the cellular roles of PhoR and AioS, and to provide a more comprehensive and specific description of bacterial adaptations to As^III^ exposure.

## 2. Materials and Methods

### 2.1. Bacterial Strains and Growth Conditions

*A. tumefaciens* 5A deletion mutants used in this study were derived using previously described cross-over PCR techniques [7] with levansucrose selection to create in-frame deletions of *phoR* and *aioS.* Growth conditions were as documented in prior reports [7,11,12]. Briefly, WT, Δ*phoR*, and Δ*aioS* strains were cultured in a defined minimal mannitol medium (MMNH_4_) overnight at 30 °C with aeration [7,13], and then centrifuged for 10 min at 3500× *g* and washed in 20 mL of 0.85% NaCl. Cells were resuspended in fresh MMNH_4_ media with 50 µM phosphate and aliquoted into ten cultures. Five of these cultures were supplemented with 100 µM As^III^ (five replicates per treatment). All cultures were incubated for six hours at 30 °C with aeration, then collected by centrifugation (10 min at 3500× *g*) and rapidly rinsed twice with 20 mL of ice cold 0.85% NaCl. Cell biomass (200 ± 5 mg per sample) was aliquoted for metabolomics and stored at −80 °C. A portion of each sample was also plated on MMNH_4_ agar plates for normalization to colony-forming units (CFU).

### 2.2. Metabolite Extraction

To extract metabolites, cells were treated as reported in Tokmina-Lukaszewska et al. [12]. Briefly, cells were lysed by two rounds of freeze-thaw in liquid nitrogen followed by sonication on ice for 5 min, and then extracted with 50% MeOH at −20 °C for 30 min. Cell lysates were centrifuged at 20,000× *g* for 15 min at −9 °C to pellet cell debris. Supernatants were centrifuged through pre-washed 100 kDa molecular weight cutoff spin filters (Pall Corporation) at 13,000× *g* for 20 min at 4 °C. Spin columns were washed twice with 100 µL 50% MeOH and centrifugation repeated. All spin column eluates were centrifuged through a pre-washed 3 kDa spin filter following the same protocol as the 100 kDa filters. The final eluates were dried using speed vacuum and stored at −80 °C until further use for metabolomics analysis.

### 2.3. NMR Analysis, Data Processing, and Statistical Procedures

Dried metabolite samples were re-suspended in 600 µL of NMR buffer (0.25 mM 4,4-dimethyl-4-silapentane-1-sulfonic acid (DSS) in 90% H_2_O/10% D_2_O, 25 mM sodium phosphate, pH 7), and transferred into 5 mm NMR tubes. All one-dimensional (1D) 1H NMR spectra were recorded at 298 K using a Bruker AVANCE III solution NMR spectrometer operating at 600.13 MHz (1H Larmor frequency) magnetic field strength. The instrument was equipped with a 5 mm liquid-helium-cooled TCI cryoprobe with Z-gradient and a SampleJet automatic sample loading system. NMR data was acquired using the Bruker-supplied 1d water suppression pulse sequence ‘noesypr1d’ with 256 transients, a 1H spectral window of 9600 Hz, 32K data points, a dwell time interval of 52 μs, and a recovery (D1) delay of 2 s between acquisitions. The NMR spectra were first processed with the Bruker TOPSPIN 3.5 software (Bruker Inc., Billerica, MA, USA) using standard parameters for referencing and applying an EM line broadening function of 0.3 Hz. The spectra were phased manually and a qfil polynomial function of 0.2 ppm in width was applied to subtract the residual water 1H NMR signal. Metabolite identification and quantification were conducted using the Chenomx v8.3 software (Chenomx Inc., Edmonton, AB, Canada) and the associated small molecule spectral reference database for 600 MHz (^1^H Larmor frequency) magnetic field strength NMR spectrometers [14]. DSS (0.25 mM) present in each sample was used as an internal reference for metabolite quantification, while the NMR signals corresponding to imidazole were used to correct for small chemical shift changes originating from slight pH variations.

Resulting lists of metabolites and concentrations normalized to CFUs were uploaded to MetaboAnalyst 4.0 [15] for univariate and multivariate statistical analysis. In MetaboAnalyst, metabolite concentrations were log-transformed and auto-scaled (mean centered divided by the standard deviation of each variable) prior to univariate and multivariate analysis. Student t-test, 2D principal component analysis (2D-PCA) and 2D partial least squares discriminant analysis (2D-PLS-DA) were performed to identify distinct metabolite patterns associated with the different bacterial strains and cell growth conditions. In addition, variable importance in projection (VIP) scores were generated from 2D-PLS-DA analyses to assess the significance of each variable (i.e., metabolite) in the projections of the 2D-PLS-DA model building [15]. Changes in metabolite levels were also used to assess which metabolite profiles contributed most to the separation of the different cellular groups in resulting 2D-PCA and 2D-PLS-DA scores plots. Metabolomics data has been deposited in the Metabolomics Workbench repository.

### 2.4. LC-MS Instrumentation, Data Acquisition, and Data Processing

LC-MS analysis was performed on an Agilent 1290 UPLC coupled to an Agilent 6538 Q-TOF mass spectrometer (Agilent Technologies, Santa Clara, CA, USA). MS was conducted in positive ion mode for hydrophilic interaction liquid chromatography (HILIC) and reverse-phase LC runs. A capillary voltage of 3500 V, fragmentation voltage of 120 V, and skimmer set at 45 V. Nitrogen drying gas (350 °C at a flow of 12 L/min and nebulizer pressure of 55 psi) were used to facilitate desolvation. Spectra were collected over a 50–1700 m/z range at a rate of 1 spectrum per second. Samples were run in randomized order with a pooled sample used for quality control (QC) which was injected at the beginning, middle, and end of the LC-MS sample queues.

Dried metabolite pellets were resuspended in 50 µL of 50% MeOH, and then chromatographic separation for polar and non-polar metabolites was achieved using two different LC columns. For polar metabolites, the cellular extract was diluted 10-fold and 10 µL was injected into a Cogent Diamond Hydride HILIC column (150 mm × 2.1 mm, 4 µm, 100 Å) (Microsolv Technology Corporation). For the HILIC column, a 25-min 99%–30%B gradient was employed using 10 mM aqueous CH_3_COONH_4_ (solvent B) and 10 mM CH_3_COONH_4_ in 95% acetonitrile (solvent A), with a 0.6 mL/min flow rate and temperature of 25 °C. As per QC runs, retention time shift was 14 s and calculated mass error was 2 ppm, with a 17% relative standard deviation of peak areas. For non-polar metabolites, 10 µL of undiluted metabolite extract was injected into a Zorbax RRHD Eclipse Plus reverse phase C18 column (150 mm × 2.1 mm, 1.8 µm) (Agilent Technologies). For the reverse phase column, a 35-min 2%–98%B gradient was employed using 0.1% formic acid in acetonitrile (solvent B) and 0.1% formic acid (solvent A). Retention time shift for the C18 column was <2 s, calculated mass error was <11 ppm, and relative standard deviation of peak areas was <7%.

For MS-MS data collection, the acquisition rate was set at 1 spectrum per second with a scan range of 50–1300 m/z (auto mode) or 50–800 (targeted mode). Isolation width was 4 m/z and collision energy set at 35 V for targeted mode or linear gradient for auto mode. Identifications of MS-MS data were made by matching fragmentation patterns to the MetLin database [16,17] Additional IDs were made using an in-house database of compounds by m/z match.

MS data acquisition, spectral analysis, and conversion of raw data files to MZxml format was performed in MassHunter (Qualitative Analysis version B.04.00, Agilent Technologies). XCMS [18] was used for detection of mass features and alignment, ran with default parameters for UPLC-Q-TOF, with the exception of peak width settings, which were modified to minimum 5 s and maximum 20 s (C18) and maximum 40 s (HILIC). Any zeros in the data (<0.4% overall) were imputed with the average of treatment group. MetaboAnalyst 4.0 [15] was used for autoscaling of data, statistical analysis, and generation of 2D-PCA plots. Metabolomics data has been deposited in the Metabolomics Workbench repository.

### 2.5. Transcriptomics Data

Gene expression data incorporated into this study originated from a recently published dataset [11]. Briefly, bacterial strains and growth conditions were the same as described above, except for 10-fold decreased iron content in the media due to iron interference with RNA extraction and purification. Iron limitation was judged not to be an issue in the cultures because the short duration of the cell culture (6 h) would not have resulted in an iron starvation scenario with a beginning iron content of 6 µM. Indeed, Rawle et al. [11] showed there was no evidence of iron limitation in the transcriptional response (40 out of 41 iron-related genes were down-regulated, the opposite of what would be expected under iron limitation).

RNA was extracted using a RNeasy^®^ Mini Kit (Qiagen Inc., Germantown, MD, USA) with DNase digestion on-column. RNA was prepped and sequenced at the Brigham Young University DNA Sequencing Center (Provo, UT, USA) utilizing the Illumina Ribo-Zero rRNA Removal Kit for ribosomal RNA depletion and the Illumina TruSeq Stranded Total RNA Sample Prep Kit for cDNA library creation. cDNA was sequenced using an Illumina HiSeq 2500 platform and raw reads were processed, normalized, and statistically analyzed using Trimmomatic [19], Kallisto [20], and R (package “Sleuth”) [20]. Only differentially regulated genes with transcript per million (TPM) > 1 (normalized transcript abundance), fold change > 2, and a q-value < 0.05 were used in the current analysis, accounting for a total of 1546 genes.

### 2.6. Pathway Annotation

NMR- and LC-MS-identified metabolites were assigned to different metabolic pathways using the topology search tool of MetaboAnalyst 4.0 [15], and groups of metabolites were classified according to the Kyoto Encyclopedia of Genes and Genomes (KEGG) pathway annotation networks [21,22]. Enzymes annotated to the same KEGG pathways were retrieved and genes from our transcriptomics dataset were matched with pathways by name and E.C. number (if available).

## 3. Results

To assess the metabolic impacts of the important regulatory kinases PhoR and AioS during As^III^ oxidation, *A. tumefaciens* 5A cells were cultured in phosphate limiting conditions to induce *aioSRBA* expression via PhoR-B regulation [7,8]. Cells were allowed to grow for six hours to ensure a full metabolic response to As^III^, as previously documented [7]. The growth conditions mirror those employed in our previous studies with wild-type *A. tumefaciens* 5A, which detailed the metabolic responses of wild-type (WT) cells exposed to As^III^ [12] and the transcriptomic responses [11]. The present study focused on parallel analyses of metabolic changes occurring in Δ*phoR* and Δ*aioS* mutant strains, utilizing ^1^H NMR and LC-MS for untargeted metabolomics analysis.

### 3.1. Metabolomics Profiles of ΔphoR and ΔaioS Mutants

^1^H NMR analysis of metabolite extracts from WT, Δ*phoR*, and Δ*aioS A. tumefaciens* cells cultured with and without As^III^ resulted in the unambiguous identification and quantification of 33 intracellular metabolites present in each sample group. To visualize the overall metabolic differences between WT and the Δ*phoR* and Δ*aioS* mutants, an unsupervised 2D principal component analysis (2D-PCA) was performed (Figure 1A,B). 2D-PCA scores plots indicated that the WT and Δ*aioS* metabolomes differed very little, irrespective of the presence or absence of As^III^. In contrast, the Δ*phoR A. tumefaciens* mutant clearly separated from that of WT and Δ*aioS* strains, both in the presence and absence of As^III^ (Figure 1A,B). This observation supports previous reports about PhoR function in phosphate limiting conditions [9,23,24], and reveals metabolic adaptations that are reflected in distinct patterns of gene expression [11]. Specifically, PhoR has a considerably larger metabolic footprint in both the absence and presence of As^III^ compared to AioS. NMR metabolite profiles were further analyzed using 2D partial least squares discriminant analysis (2D-PLS-DA) to identify metabolites whose change in abundance contribute most to the separation between the *A. tumefaciens* 5A WT, Δ*aioS*, and Δ*phoR* groups, as determined by variable importance in projection (VIP) scores (Appendix A). Only metabolites with VIP values greater than one were considered to be significant [15], and used in subsequent metabolic pathway impact analysis.

To extend metabolome coverage, untargeted LC-MS analysis was performed using both reverse-phase (RP) and HILIC chromatography. In total, 3092 non-polar (RP) and 1010 polar (HILIC) features were detected across all samples (Appendix A). The larger number of non-polar features is consistent with trends observed in previous metabolomics analyses of WT cells [12]. Of the detected LC-MS features, 23 metabolites were identified by accurate mass and fragmentation pattern (MS-MS). An additional 18 were identified using an in-house standard database, with five metabolites identified using both methods. 2D-PCA was performed on all MS features to display separation patterns between groups (Figure 1C–F). In the presence of As^III^ (Figure 1D,F), WT and Δ*aioS* profiles were more similar to each other, whereas Δ*phoR* was more distinct; these patterns are similar to those identified by NMR (Figure 1B). In the absence of As^III^, no separation between the three different *A. tumefaciens* cell types is observed, as assessed by the non-polar metabolite profiles (Figure 1E). This finding indicates that these pools of non-polar metabolites are not heavily impacted by the *aioS* and *phoR* mutations. However, a distinct separation between WT and Δ*aioS* was observed based on differential polar metabolite profiles in the absence of As^III^ (Figure 1C), and less so in the presence of As^III^ (Figure 1D). Differences between WT and Δ*aioS* cells in the presence of As^III^ were most apparent in the non-polar metabolite fraction (Figure 1F). These patterns suggest that in the absence of As^III^, polar metabolite pools are more affected by the loss of AioS function, whereas in the presence of As^III^, non-polar metabolites are more impacted by loss of AioS. With respect to PhoR, the non-polar metabolite pool was the least sensitive to the Δ*phoR* mutation in the absence of As^III^ (Figure 1E), while all other comparisons resulted in distinct Δ*phoR* separation from the other cell types (Figure 1A–D,F).

Pairwise comparisons of metabolite profiles were examined between WT and Δ*aioS* and Δ*phoR* mutants using the ^1^H NMR and LC-MS metabolomics data. Among identified metabolites, 37 exhibited significantly different levels between sample groups (Table 1). These metabolites included amino acids (Ala, Pro, Val, Trp, Tyr, Arg, Leu, Ile, Lys), sugars (ribose, sucrose, maltose, maltohexose, maltotetraose, maltopentaose), and other key metabolic indicators of cell function (betaine, choline, cytosine, adenosine, putrescine, nicotinate). Most fold changes were in the 1.5–3 range, though some were as high as 10-fold (betaine, sorbitol). Comparing differences between the strains highlighted specific metabolic patterns and regulatory networks involving PhoR and AioS that include: (i) metabolites variably affected by both histidine kinases regardless of As^III^ (e.g., β-alanine, betaine) or exclusively in the presence of As^III^ (e.g., arginine, glutamate); (ii) metabolites affected only by PhoR, whether in the absence of As^III^ (e.g., maltose, dipeptide Ala-Gly), presence of As^III^ (e.g., 5-oxoproline, isonicotinate), or both (e.g., cytosine, glutamine) and iii) metabolites affected only by AioS, whether in the absence of As^III^ (e.g., hypoxanthine) or only the presence of As^III^ (e.g., maltopentose). Similar to the information visualized in 2D-PCA scores plots (Figure 1), changes in these metabolite abundance patterns (Table 1) report on the distinct metabolomes of Δ*phoR* and Δ*aioS*, and detail more specifically which metabolites are affected by either PhoR or AioS, or both.

### 3.2. Pathway Analysis Using Transcriptomics and Metabolomics Data

To interpret the metabolomics results within a most up-to-date context, we integrated the new findings on the Δ*phoR* and Δ*aioS* metabolomes (Table 1, Figure 1) with our recent WT *A. tumefaciens* transcriptomics and metabolomics data [11,12]. To accomplish this, transcripts and metabolites were classified according to KEGG pathway designations to survey whether a cellular pathway was influenced at either or both omics levels. A full detailed list of impacted metabolic pathways in the WT and mutants is included in Appendix A and summarized in Figure 2. Important caveats to consider in these pathway analyses are that some metabolites (e.g., glutamate) are common to numerous KEGG pathways and thus linking them to specific genes/functions is challenging. Furthermore, it is important to keep in mind that changes in gene expression do not always occur on the same timescale as changes in metabolite levels; thus, at a single time point after perturbation, a one-to-one correspondence between transcriptomics and metabolomics changes is not always expected. Below we examine these pathways not to overstate correlations between transcriptomics and metabolomics data, but rather to survey the evidence of potential cellular networks impacted by As^III^ exposure and regulated by PhoR and/or AioS in these strains.

At a global level of analysis in the wildtype strain, perturbations of gene transcription and metabolite levels were apparent for a number of KEGG pathways (Figure 2, Appendix A). In comparing the Δ*phoR* and Δ*aioS* mutants to WT, seven KEGG pathways were found to be perturbed as a result of As^III^ exposure in both mutants at both transcriptional and metabolic levels (Figure 2A, Appendix A). These pathway classifications mirrored metabolite abundance trends (Figure 1, Table 1) where PhoR consistently had a larger influence over metabolism than AioS. In some cases, PhoR influence on cellular pathways involved metabolic networks that were not impacted by As^III^. For example, when comparing the Δ*phoR* mutant to the WT with respect to arginine and proline metabolism (Appendix A), down-regulation of genes encoding homospermidine synthase (AT5A_02715), proline dipeptidase (AT5A_23006), and a spermidine/putrescine transporter (AT5A_20446) matched the observation of reduced levels of proline and putrescine, irrespective of As^III^ exposure. Within this same pathway category however, changes in several gene transcripts and relevant metabolites were observed in the Δ*phoR* mutant but only in the presence of As^III^. Examples included genes coding for arginase, an arginine biosynthesis bifunctional protein (*argJ*), and ornithine cyclodeaminase, where transcript expression patterns correlated with observed altered levels of arginine and glutamate (Appendix A). AioS seemed to play a much smaller role, only affecting the expression of one or two genes and/or metabolite levels in each pathway during As^III^ exposure (Figure 2A, Appendix A). Specifically, decreased transcript levels of ornithine cyclodeaminase and decreased levels of arginine and glutamate were the only changes observed in the arginine and proline metabolism cluster for the Δ*aioS* mutant compared to the WT in the presence of As^III^ (Appendix A).

Other cellular pathways impacted by As^III^ in both Δ*aioS* and Δ*phoR* mutants included glutathione metabolism, pantothenate and coenzyme A metabolism, galactose metabolism, fructose and mannose metabolism, and valine, leucine, and isoleucine metabolism (Figure 2A). Again however, the influence of AioS was much less extensive than PhoR. Other pathway designations impacted exclusively by PhoR (Figure 2B) included phenylalanine and nicotinate/nicotinamide metabolism, suggesting these responses do not directly involve AioS.

Several pathways were affected by PhoR at both transcriptomics and metabolomics levels, but only at one or the other of the omics levels by AioS (Figure 2C,D). These included five pathways that were influenced at the gene level in both mutants and at the metabolite level by PhoR but not AioS (Figure 2C). These networks were associated with changes in gene transcripts/metabolites involved in the pentose phosphate pathway, sucrose/starch metabolism, glycolysis/gluconeogenesis, purine metabolism, and glycerophospholipid metabolism. It appears that the few AioS-regulated genes differentially expressed in these categories did not have a direct effect on metabolite pools, at least at the bacterial cell growth time point sampled in this study (Appendix A). Pathways with Δ*aioS* and Δ*phoR* perturbation at the metabolite level, but with gene expression changes only in the Δ*phoR* mutant (Figure 2D), included the shikimate pathway, and metabolism of glyoxylate, pyruvate, taurine/hypotaurine, pyrimidines, glutamate/glutamine, alanine, and aspartate. These patterns suggest that the influence of AioS on metabolite levels may be due to AioS-based gene regulation further up or downstream of the relevant pathways, or as a result of gene regulation that was not captured in the transcriptomics data obtained after six hours of cell response to As^III^.

### 3.3. Multi-Omics Mapping of Carbon Metabolism during As^III^ Exposure

To focus on important controls affecting cell metabolism during As^III^ exposure, the metabolomics and transcriptomics data were integrated into a detailed model of potential carbon flow taking into account the regulatory influences of PhoR and AioS during As^III^ oxidation (Figure 3, Figure 4 and Figure 5, Appendix A). Carbon flow begins with mannitol, the sole carbon source in the minimal media. This model builds upon previous work which detailed metabolic bottlenecks in carbon metabolism as a result of As^III^ inhibition of pyruvate dehydrogenase (PDH) [25] and α-ketoglutarate dehydrogenase (KGDH) [26].These inhibitory blocks lead to metabolic diversions stemming from pyruvate and α-ketoglutarate [12], as well as the build-up of hypoxanthine resulting from xanthine oxidase inactivation by As^III^. These bottlenecks presumably arose from post-translational enzyme inactivation and thus would not necessarily be correlated with a transcriptional response.

To make comparisons with the mutants, we first examined the metabolomics and transcriptomics model for the WT +/- As^III^ (Figure 3). The data continue to support the concept of metabolic diversion due to protein inactivation. No genes that were differentially expressed were found that directly affect the levels of xanthine and hypoxanthine, suggesting that their change in abundance upon As^III^ exposure is due to the known As^III^ inactivation of xanthine oxidase [27,28], as previously hypothesized [12]. For the metabolite diversions at the PDH and KGDH reaction steps, several relevant genes were found to be differentially expressed, but they did not seem to exert a strong influence on overall metabolite levels. For example, the gene encoding dihydroxy-acid dehydratase (AT5A_22266) was decreased 2.2-fold during As^III^ exposure. This enzyme is involved in valine and isoleucine production from pyruvate; however, valine and isoleucine levels were increased following As^III^ exposure in the WT (Figure 3, Appendix A). Potential reasons for apparent differences between gene expression and associated metabolite levels include: (i) the change in mRNA level does not significantly impact protein level, and/or (ii) there exists a temporal shift between transcription and associated metabolic changes that is not captured in this single sampling time point of cellular growth and response to As^III^. By comparison, an example of metabolite change and transcript expression that were consistent with each other, included increased levels of ornithine and putrescine, with concurrent increase in transcript expression of ornithine decarboxylase (AT5A_00120), which may account for the observed increased putrescine levels. Furthermore, transcription of genes coding for putrescine transporters was decreased, which could restrict putrescine trafficking in and out of the cell. Overall however, metabolite abundances in the WT were increased irrespective of transcript up- or down-regulation and this pattern seems to be more strongly associated with As^III^ inactivation of key enzymes rather than transcriptional control affecting the metabolic flow through these pathways (Figure 3).

Considering the same model of carbon metabolism discussed above (Figure 3), regulatory impacts of the Δ*aioS* mutation appeared very limited when compared to the WT grown in the presence of As^III^ (Figure 4), affecting only 13 metabolites and four genes (though none of the affected genes were the same as those affected in the WT upon As^III^ exposure, (Figure 3). Almost all affected metabolites and genes in the Δ*aioS* mutant were decreased in abundance as compared to the WT, implying that AioS-based signaling in the WT cells would normally have an enhancement effect. Focusing on the enzyme blockages, no change in hypoxanthine and xanthine levels were observed in the Δ*aioS* mutant grown in the presence of As^III^, nor were any associated transcript levels altered, suggesting that the changes in the WT (Figure 3) are due to As^III^ inactivation of xanthine oxidase and not Δ*aioS* influence(s). Regarding the PDH and KGDH enzyme blockages, however, AioS appeared important for production of valine and lactate, and glutamate and arginine, respectively, although transcriptional influence was minimal. Expression of two genes, encoding enzymes involved in the breakdown of valine (AT5A_1590, 1595), may have had influence over valine levels, which were decreased in the Δ*aioS* mutant, and decreased expression of transcripts encoding an ornithine cyclodeaminase (AT5A_17276) could have contributed to the observed decrease in arginine levels. The few other metabolites that were decreased in abundance in the Δ*aioS* mutant (β-alanine, tryptophan, raffinose, maltotriose, and maltopentose) were not associated with any direct observed transcriptional change in obviously relevant genes, at least according to our current gene annotations.

The Δ*phoR* mutation had a much larger influence on metabolism with regard to both transcription and metabolites (Figure 5). Two genes (AT5A_16821, 09485) and the majority of metabolites that were differentially expressed in the WT during As^III^ exposure (Figure 3), were further impacted by PhoR (Figure 5), as well as an additional 26 genes, indicating the necessity of PhoR for normal metabolic function during As^III^ exposure. One exception corresponded to hypoxanthine metabolism, where xanthine and hypoxanthine levels appeared unaffected by the Δ*phoR* mutation (as well as Δ*aioS*, Figure 4) as compared to WT levels. This again indicates that As^III^ inactivation of xanthine oxidase is the primary source of altered levels of these metabolites (Figure 3). Irrespective of the presence of As^III^, metabolites altered in the Δ*phoR* mutant (but not Δ*aioS*) included isoleucine, leucine, lactate, valine, nicotinate, ribose, and sucrose. At the transcriptional level, some gene expression patterns seemed to be altered either in the absence (Appendix A) or presence of As^III^ (Figure 5), but few genes were affected in both conditions (Figure 5). Many perturbed genes in the Δ*phoR* mutant grown in the presence As^III^ encode functions centered around the PDH blockage, particularly genes encoding enzymes for the catabolism of leucine, isoleucine, and valine to acetyl-CoA. However, the levels of the corresponding metabolites were unaltered by As^III^ exposure (rather, just the Δ*phoR* mutation). Thus, it appears that the PhoR impact over transcriptional expression did not translate into a significant effect on the levels of those metabolites when the Δ*phoR* mutant is grown in the presence of As^III^. In contrast, at the KGDH blockage, most of the metabolite and transcriptional changes affected in the Δ*phoR* mutant were linked to As^III^ exposure. Clearly, perturbation of metabolite and transcript levels demonstrate that PhoR influences the metabolic flow of glutamate during As^III^ exposure (Figure 5). When viewed at a broad perspective, the trends indicate that PhoR has a significant impact on carbon metabolism during As^III^ exposure.

## 4. Discussion

Research on As^III^-resistant organisms has characterized various functions induced by As^III^ exposure [11,29,30,31,32,33], ranging from direct arsenic responses like arsenic resistance (*ars* genes), As^III^ oxidation and oxidative stress, to general cell functions including remodeling of carbon and amino acid metabolic pathways. There are several underlying factors that influence these metabolic changes during As^III^ exposure and, as we document for *A. tumefaciens* 5A, can be quite complex. These factors include: (1) protein inactivation by As^III^; (2) PhoR- and AioS-based regulation and (3) influences of other As^III^-responsive systems (e.g., *ars* genes). To clearly highlight factors driving cell metabolism during As^III^ oxidation, each will be discussed in turn with a focus on carbon metabolism.

### 4.1. Protein Inactivation by As^III^

As^III^ inactivation of specific enzymes is well documented [34] and is undoubtedly a major factor impacting the metabolite patterns observed in our data. There are enzymes where inhibition by As^III^ is well characterized and as such offer an opportunity to directly assess whether there are other layers of cell response(s) at play. Specifically, is the altered carbon flow inferred from the metabolomics studies due solely to enzyme inhibition or is there evidence that the cell response is more direct and organized? The advanced status of our understanding of the regulatory systems governing As^III^ responses in *A. tumefaciens* 5A provides a good opportunity to examine this directly, although with some caveats (discussed below).

One example is xanthine oxidase, which is inactivated as a result of As^III^ interaction with the enzyme molybdenum cofactor [28,35]. Other similarly affected key cellular proteins include pyruvate dehydrogenase (PDH), α-ketoglutarate dehydrogenase (KGDH), and branched-chain alpha-ketoacid dehydrogenase complexes, which require a dihydrolipoamide subunit that is deactivated by As^III^ [36,37]. The resulting enzyme dysfunction leads to repurposing of the carbon metabolic pathways without necessarily invoking gene transcriptional changes [12,32]. This pattern of As^III^ inactivation was evident for xanthine oxidase, where levels of xanthine and hypoxanthine were altered in the WT + As^III^ group (Figure 3, Appendix A), but without any transcriptional influence in the WT (Figure 3) or in the mutants in the presence of As^III^ (Figure 4 and Figure 5). With respect to the pathway blocks at PDH and KGDH however, the results are more complex to interpret. Formate, malate, and fumarate were the only metabolites whose levels were impacted in the WT without any associated WT transcriptional changes (Figure 3) or mutant effects (Figure 4 and Figure 5), and thus seem to be impacted mainly by post-translational As^III^-inactivation of PDH or KGDH. Levels of other metabolites stemming from pyruvate and α-ketoglutarate metabolism, however, appear to be additionally influenced by other factors because transcriptional and/or metabolite alterations were observed in both mutant strains (Figure 4 and Figure 5, Appendix A).

The WT data do not provide strong evidence for a transcriptional influence over genes that encode metabolic enzymes which could be used to bypass the PDH or KGDH bottlenecks, as the small amount of WT transcriptional changes did not translate into significant changes in metabolite levels (Figure 3). However, the importance of phosphoenolpyruvate (PEP) carboxykinase (AT5A_17576) in committing cell metabolism to gluconeogenesis could indicate an important step that directs cell metabolism towards production of sugars (maltose, ribose, raffinose, stachyose) (Figure 3), or to support metabolic flow into the shikimate pathway (as evidenced by the clear patterns observed in the data shown in Figure 3). As^III^ influence over levels of ornithine and putrescine, and the relevant enzymes ornithine decarboxylase and putrescine transporters in the WT (Figure 3), could indicate a way to increase intracellular putrescine as a mechanism of stress management. Putrescine concentrations have been positively correlated with growth and play a role in stimulating transcriptional responses under stress, where cells with impaired putrescine metabolism display defective stress responses [25].

### 4.2. PhoR- and AioS-Based Regulation

In addition to As^III^-inactivation of key enzymes, PhoR and AioS regulatory controls impact cellular metabolism in *A. tumefaciens* 5A during As^III^ exposure [7,8,11]. PhoR impacts on metabolic profiles were evident across the board, while AioS’s influence was considerably smaller, consistent with published transcriptomics data [11] (Figure 2, Figure 4 and Figure 5, Appendix A). By mapping identified metabolites in the context of our data on the regulation during As^III^ exposure in *A. tumefaciens* 5A, we were able to incorporate the data within the most current framework of impacted networks (Figure 2, Figure 3, Figure 4 and Figure 5). Considering carbon metabolism as an example, PhoR and AioS both influence metabolic flow. At the PDH block, for example, expression patterns for genes encoding functions that facilitate the conversion of pyruvate to various amino acids indicated that PhoR has a greater impact on cellular functions in the presence of As^III^ (Figure 5), although the changes in associated metabolite levels were not As^III^-specific (Figure 5) and the transcriptional changes, by themselves, would indicate that carbon flow through these metabolites would be reduced. On the other hand, AioS affected several metabolites in the presence of As^III^, but not in conjunction with transcriptional changes. There were however, tighter correlations of relevant mRNA and metabolite levels at the KGDH-catalyzed reaction step where both metabolite and transcriptional changes in the Δ*phoR* mutant were observed in the presence of As^III^. It is evident that AioS and PhoR impact carbon metabolism because of altered metabolite and/or gene transcription levels in the mutants, although correlation between the two omics levels does not always offer clear insights into the mechanism. Furthermore, this suggests that there may be other uncharacterized factors at play (discussed further below).

As seen in other multi-omics datasets, increases or decreases in the abundance of transcripts and metabolites is not always correlated [38,39,40]. Moreover, gene transcription changes do not account for post-translational modifications nor is metabolite flow necessarily directly correlated with steady state metabolite levels [40]. Therefore, viewing network connectivity and perturbation as a whole is important for understanding the biological significance of metabolite level changes identified from omics data [41,42], as only studies detailing temporal changes in mRNA and metabolite fluxomics would be able to directly make these types of correlations. As such, the consistency of pathway-level perturbations inferred from mRNA and metabolite levels in the Δ*phoR* and Δ*aioS* mutants (Figure 2, Appendix A) demonstrates that As^III^, PhoR, and AioS (to a lesser extent) are important regulators of global metabolism during a transition phase where cells prepare themselves to cope with the toxic effects of As^III^, in addition to inducing As^III^ oxidation.

### 4.3. Influences of Other As^III^-responsive Systems

A third level of metabolic regulation likely occurs through other transcriptional regulators. PhoR is known to regulate a considerable number of transcriptional regulators (~50 in *A. tumefaciens* 5A) [11], and some of the metabolic perturbations in our study are undoubtedly the result of downstream signaling mediated by these proteins. Other regulatory impacts observed in the strains could be due to the ArsR proteins, which are As^III^-sensitive transcriptional regulators that control arsenic-microbial interactions. Traditionally these proteins have been characterized as classic repressors; however, recent studies indicate that ArsR proteins in *A. tumefaciens* 5A have both repressor and activator activity [43] over a variety of cell functions in addition to arsenic resistance. We have documented that AioS impacts transcriptional expression of two of these proteins, ArsR2 and ArsR4 in the presence of As^III^, and there are also two uncharacterized ArsR family regulators impacted in the mutants (one by AioS, one by PhoR) [11]. The metabolic footprint of these regulators is likely another important factor contributing to the global cell regulation during As^III^ exposure in *A. tumefaciens* 5A. Additionally, in both the Δ*phoR* and Δ*aioS* mutants under As^III^ exposure, transcriptional responses for a considerable number of uncharacterized proteins (almost 100 in Δ*phoR* vs. WT, 21 in Δ*aioS* vs. WT) was documented [11], and it would not be unreasonable to suggest that one or more of these proteins impact the metabolic responses documented in this study.

As a final consideration, even though prior work showed full induction of As^III^ oxidation at six hours under phosphate limiting conditions [7], it is at least possible that a full transition to an environment with As^III^ may take longer, and that the cells harvested at six hours were still in the midst of adjusting their overall metabolic response. Assessing later time points would provide evidence as to whether the apparent uncoupling of gene and metabolite expression is a result of cells being in a transitional metabolic state, and/or simply a confounding factor of sampling metabolism at a single time point of cellular growth and As^III^ exposure.

In summary, PhoR and AioS are important regulators that govern metabolic responses in *A. tumefaciens* 5A during As^III^ exposure. Transcriptional and metabolic profiles of the Δ*phoR* and Δ*aioS* mutants demonstrated a large contingent of cell functions affected by PhoR, but a considerably smaller number affected by AioS. In addition to documented arsenic-specific responses like As^III^ oxidation and arsenic resistance, PhoR and AioS were shown to influence fundamental cell functions under As^III^-PSR conditions, including carbon, amino acid, and sugar metabolism. This study has provided metabolic profiles detailing PhoR and AioS influence, linked these data to generate the most up-to-date framework for understanding the role of these key regulators, and provided a starting point for investigating key metabolic changes unexplained by current protein annotations in *A. tumefaciens* 5A. These insights indicated metabolic networks that respond to As^III^ exposure and highlight the impact that As^III^-oxidizing microbes likely have on key biogeochemical cycles in ecological systems.

## Figures and Tables

**Figure 1 microorganisms-08-01339-f001:**
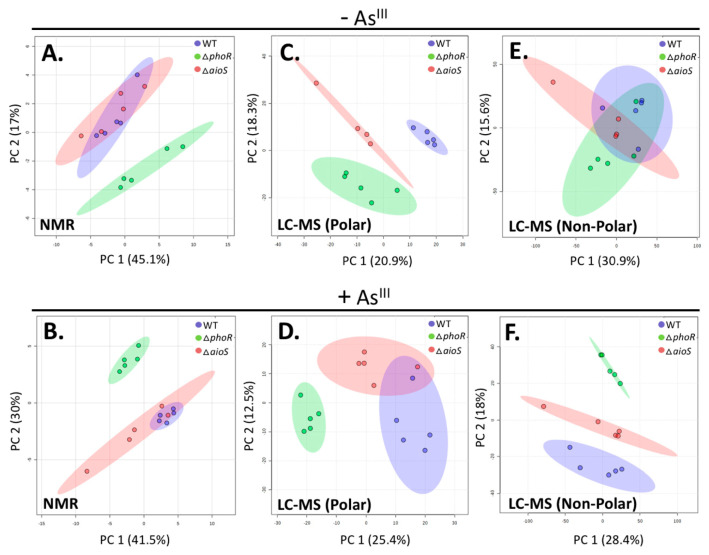
2D Principal Component Analysis (i.e., 2D-PCA scores plots) of metabolite profiles from ^1^H nuclear magnetic resonance (NMR) and untargeted liquid chromatography-mass spectrometry (LC-MS) metabolomics data. Metabolites from cultures grown without As^III^: (**A**) NMR, (**C**) polar LC-MS, and (**E**) non-polar LC-MS metabolites. Metabolites from cultures grown in the presence of As^III^: (**B**) NMR, (**D**) polar LC-MS, and (**F**) non-polar LC-MS. Wild-type (WT) = purple; Δ*phoR* = green; Δ*aioS* = red. Shaded ellipses denote 95% confidence intervals.

**Figure 2 microorganisms-08-01339-f002:**
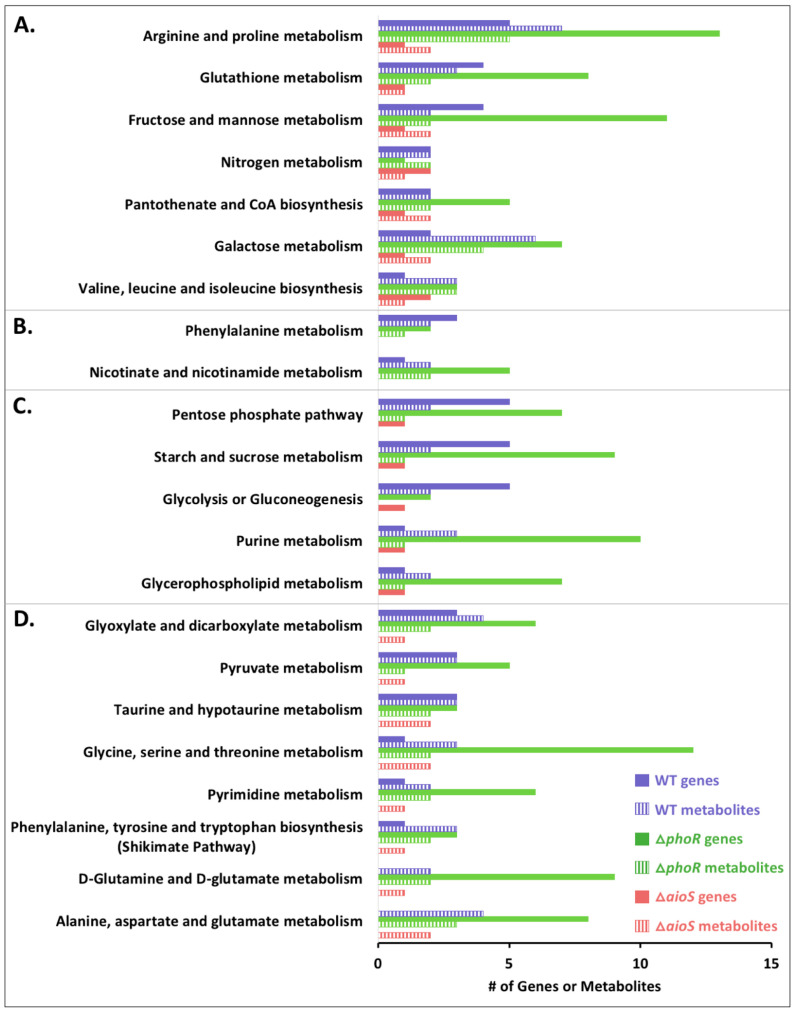
Cellular functions perturbed by As^III^ in WT and mutants, which contribute to observed differences between treatments. Only genes and metabolites that were different in pairwise comparisons (WT +/- As^III^; mutant + As^III^ vs. WT + As^III^) are shown. Data was classified by KEGG pathways. (**A**) Pathways perturbed at the transcript and metabolite levels in the WT + As^III^ and in both mutants + As^III^. (**B**) Pathways altered at the transcript and metabolite levels in the WT + As^III^ and in the Δ*phoR* mutant + As^III^. (**C**) Pathways altered at both the transcript and metabolite levels in the WT + As^III^ and the Δ*phoR* mutant + As^III^, but only at the transcript level in the Δ*aioS* mutant + As^III^. (**D**). Pathways regulated in the WT + As^III^, both at the transcript and metabolite levels in the *ΔphoR* mutant + As^III^, but only at the metabolite level in the Δ*aioS* mutant + As^III^.

**Figure 3 microorganisms-08-01339-f003:**
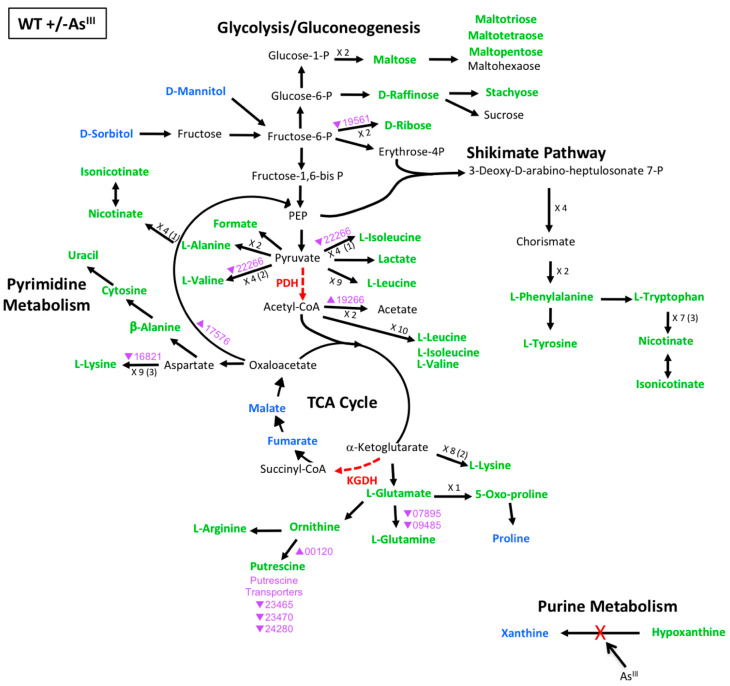
Model of carbon metabolism in the WT during As^III^ exposure, with mannitol being the initial substrate. Reaction steps were derived from KEGG pathway maps using the most parsimonious routes of metabolite formation, noting that not all enzyme reaction steps are depicted in the model. When multiple intermediates are involved between illustrated metabolites, the number of reactions is shown; i.e., ×3, with the number of intermediates identified shown in parentheses. Green text denotes metabolites increased in abundance; blue text denotes metabolites decreased in abundance. Red dashed vector arrows indicate reactions suggested to be inhibited by As^III^; (PDH = pyruvate dehydrogenase; KDGH = alpha-ketoglutarate dehydrogenase). Transcripts are denoted by AT5A identification number (purple text), with a triangle indication increased expression (←) or decreased expression (↔) upon As^III^ exposure.

**Figure 4 microorganisms-08-01339-f004:**
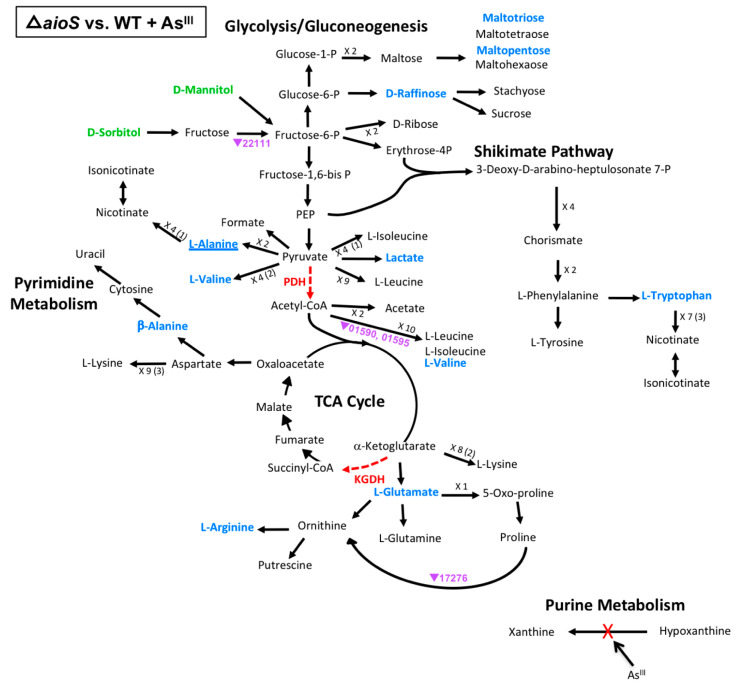
Model of carbon metabolism in the Δ*aioS* mutant vs WT during As^III^ exposure, with mannitol being the initial substrate. Reaction steps were derived from KEGG pathway maps using the most parsimonious routes of metabolite formation, noting that not all enzyme reaction steps are depicted in the model. When multiple intermediates are involved between illustrated metabolites, the number of reactions is shown; i.e., ×3, with the number of intermediates identified shown in parentheses. Green text denotes metabolites increased in abundance; blue text denotes metabolites decreased in abundance. Red dashed vector arrows indicate reactions suggested to be inhibited by As^III^. Transcripts are denoted by AT5A identification number (purple text), with a triangle indication increased expression (←) or decreased expression (↔). Underlined text indicates metabolites or transcripts for which the change in abundance when compared to the WT was the same with or without As^III^.

**Figure 5 microorganisms-08-01339-f005:**
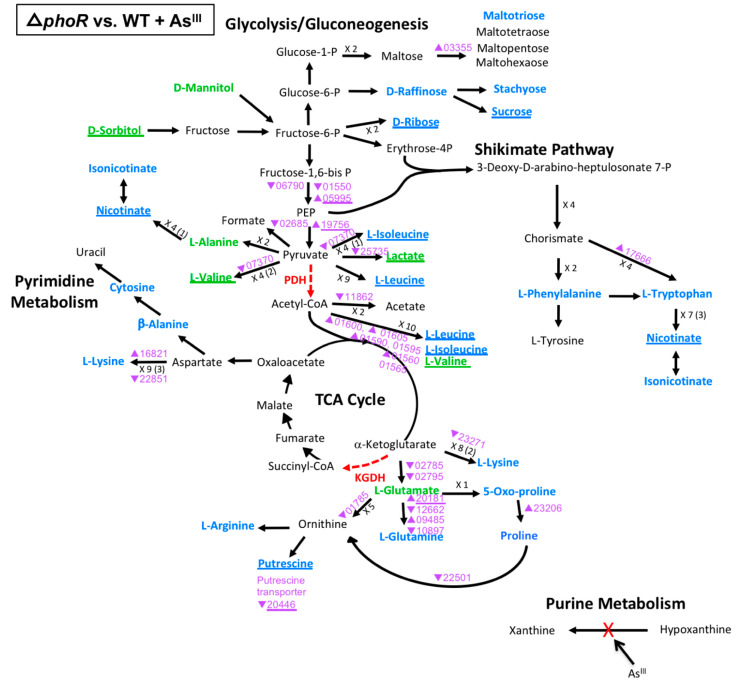
Model of carbon metabolism in the Δ*phoR* mutant vs WT during As^III^ exposure, with mannitol being the initial substrate. Reaction steps were derived from KEGG pathway maps using the most parsimonious routes of metabolite formation, noting that not all enzyme reaction steps are depicted in the model. When multiple intermediates are involved between illustrated metabolites, the number of reactions is shown; i.e., ×3, with the number of intermediates identified shown in parentheses. Green text denotes metabolites increased in abundance; blue text denotes metabolites decreased in abundance. Red dashed vector arrows indicate reactions suggested to be inhibited by AsIII. Transcripts are denoted by AT5A identification number (purple text), with a triangle indication increased expression (←) or decreased expression (↔). Underlined text indicates metabolites or transcripts for which the change in abundance when compared to the WT was the same with or without AsIII.

**Table 1 microorganisms-08-01339-t001:** Metabolites identified by liquid chromatography-mass spectrometry (LC-MS) and ^1^H nuclear magnetic resonance (NMR) in wild-type (WT) and mutants, with data represented as fold changes. Only fold changes associated with a p-value ≤ 0.05 are listed, unless otherwise noted. Metabolites are classified by regulators that appear to be involved in expression (PhoR and/or AioS). STD = LC-MS identification by authentic standard database (MS data).

Metabolite	ID Method	WT(+As)/WT(-As)	△*phoR*/WT		△*aioS*/WT		Regulation
			No As^III^	+ As^III^	No As^III^	+ As^III^	Genes Involved
Beta-Alanine	NMR	1.5	−1.2	−2.0	1.1	−1.3	*PhoR, AioS*
Betaine	MS-MS	−4.6	−6.3	−10.2		−2.0	*PhoR, AioS*
D-Mannosamine ^a^	STD	−1.5	2.6	4.1	3.4 *		*PhoR, AioS*
D-sorbitol	MS-MS	−1.5	9.8	10.6	8.5	4.7	*PhoR, AioS*
L-Alanine	NMR, MS-MS	1.3	−1.1	1.2	−1.1	−1.2	*PhoR, AioS*
L-Proline	MS-MS, STD	−1.4 *	−3.4	−2.2	−2.3		*PhoR, AioS*
L-Valine	NMR	2.2	1.6	1.4	1.1	−1.3	*PhoR, AioS*
Lactate	NMR	2.0	2.0	1.5	1.3	−1.2	*PhoR, AioS*
Maltotriose	MS-MS	2.1		−4.6	−3.9	−2.0	*PhoR, AioS*
Mannitol	NMR	−1.2 *	3.7	10.8	−2.1	1.8	*PhoR, AioS*
Sucrose	MS-MS	1 *	−2.7	−3.4	−1.7		*PhoR, AioS*
Adenosine ^b^	STD		−7.7	−6.0	−2.3		*PhoR, AioS*
Palatinose	STD			−3.1	−2.0		*PhoR, AioS*
L-Arginine	MS-MS, STD	1.1 *		−2.1		−2.3	*PhoR, AioS*
L-Glutamate	MS-MS, NMR	1.8		2.9		−1.8	*PhoR, AioS*
L-Tryptophan	MS-MS	1.4 *		−1.9		−1.6	*PhoR, AioS*
D-Raffinose ^c^	STD	2.9 *		−6.2		−4.9	*PhoR, AioS*
Cytosine	MS-MS, NMR	2.4	−2.0	−1.7			*PhoR*
Glycerophosphocholine	MS-MS	−1.6	2.1	3.5			*PhoR*
L-Glutamine	STD, NMR	3.3	−2.2	−3.1			*PhoR*
L-Isoleucine	NMR	1.6	−1.2	−1.5			*PhoR*
L-Leucine	NMR	1.7	−1.4	−1.3			*PhoR*
L-Phenylalanine	NMR, MS-MS	1.5	−1.2	−2.9			*PhoR*
Nicotinate	NMR	1.3	−1.4	−1.3			*PhoR*
Putrescine	NMR	1.5	−1.6	−1.1			*PhoR*
Ribose	NMR, MS-MS, STD	1.3	−1.4	−1.4			*PhoR*
Maltose	NMR, STD	1.3 *	1.8				*PhoR*
Ala-Gly	STD		−4.8				*PhoR*
5-oxoproline	MS-MS	1.8		−2.7			*PhoR*
Isonicotinate	MS-MS	2.4		−1.7			*PhoR*
L-Lysine	MS-MS, NMR, STD	1.5		−1.7			*PhoR*
Stachyose	STD	1.8		−1.7 *			*PhoR*
Oxypurinol	NMR	3.4			6.1	1.1	*AioS*
Hypoxanthine	MS-MS, STD	8.0 *			6.3		*AioS*
Maltohexaose	MS-MS	1.1 *			1.7		*AioS*
Maltotetraose	MS-MS	1.9			1.6		*AioS*
Maltopentaose	MS-MS	1.5 *				−1.7	*AioS*

* Fold change associated with a p-value > 0.05. ^a^ additional ID: D-Galactosamine. ^b^ additional ID: 2′-Deoxyguanosine. ^c^ additional ID: D-Melezitose.

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
