# Peer review of "Metabolic Responses to Arsenite Exposure Regulated through Histidine Kinases PhoR and AioS in Agrobacterium tumefaciens 5A"

_microorganisms, 2020, doi:10.3390/microorganisms8091339_

Round 1

Reviewer 1 Report

The manuscript by Rawle et al describes the metabolic response to arsenite exposure regulated through histidine kinases PhoR and AioS in A.tumefaciens 5A. The manuscript is generally well written but is overly long in some sections which detracts from the work and impacts the manuscripts readability. The authors disclose that they have previous transcriptomics and metabolomics data that this manuscript is building upon, which raises questions about novelty and what presented here is new information. As such, I recommended the authors undertake a re-write, cutting back a lot of the text that is contextual in previous published papers of their work and focus on the new data, the importance of the work in this manuscript and what impact it will have (what is going to change as a result of these findings).

Additional comments to consider:

  • Ensure all acronyms in figures are defined in the figure captions.
  • Ensure the figure font is of enough size for the reader to interpret data points clearly.
  • Provide more detail on the LC methods; were the samples run in both positive and negative mode? Were pooled PBQCs used? QC standards?
  • How does this work relate to the MSI?

Reviewer 2 Report

Rawle et al describe metabolic changes of Agrobacterium strain 5A to arsenite exposure. They link metabolomics data to previous transcriptomic analysis to obtain insight in the cell's response to arsenite in both wildtype and phoR and aioS mutants. The results are interesting and form the start of further detailed studies of the response and consequences of exposure to arsenite.

One critical remark from my side is that the metabolomic analysis is restricted to 33 identified compounds, but this has to be accepted as this is about the current state-of-the-art for such analyses. For technical reasons mostly limited to 50 or less compounds that can be identified. The term 'metabolomics' therefore does not yet live up to its name.
